# *Trichoderma harzianum* Volatile Organic Compounds Regulated by the THCTF1 Transcription Factor Are Involved in Antifungal Activity and Beneficial Plant Responses

**DOI:** 10.3390/jof9060654

**Published:** 2023-06-11

**Authors:** María Belén Rubio, Maurilia Maria Monti, Liberata Gualtieri, Michelina Ruocco, Rosa Hermosa, Enrique Monte

**Affiliations:** 1Department of Microbiology and Genetics, Institute for Agribiotechnology Research (CIALE), University of Salamanca, Campus de Villamayor, C/Duero, 12, 37185 Salamanca, Spain; rhp@usal.es (R.H.); emv@usal.es (E.M.); 2Institute for Sustainable Plant Protection (CNR-IPSP), Piazzale Enrico Fermi 1, 80055 Naples, Italy; liberata.gualtieri@ipsp.cnr.it (L.G.); michelina.ruocco@ipsp.cnr.it (M.R.)

**Keywords:** VOCs, volatilome, PTR-Qi-TOF-MS, gene disruption, methyltransferase, *Botrytis cinerea*, plant growth promotion, JA, SA, plant immunity

## Abstract

The transcription factor THCTF1 from *Trichoderma harzianum*, previously linked to the production of 6-pentyl-2*H*-pyran-2-one (6-PP) derivatives and antifungal activity against *Fusarium oxysporum*, has been related in this study to conidiation, production of an array of volatile organic compounds (VOCs) and expression of methyltransferase genes. VOCs emitted by three *T. harzianum* strains (wild type T34, transformant ΔD1-38 that is disrupted in the *Thctf1* gene encoding the transcription factor THCTF1, and ectopic integration transformant ΔJ3-16) were characterized by Proton Transfer Reaction-Quadrupole interface-Time-Of-Flight-Mass Spectrometry (PTR-Qi-TOF-MS). *Thctf1* disruption affected the production of numerous VOCs such as the antifungal volatiles 2-pentyl furan and benzaldehyde which were under-emitted, and acetoine, a plant systemic defense inductor, which was over-emitted. Biological assays show that VOCs regulated by THCTF1 are involved in the *T. harzianum* antifungal activity against *Botrytis cinerea* and in the beneficial effects leading to *Arabidopsis* plant development. The VOC blend from the disruptant ΔD1-38: (i) inhibited *Arabidopsis* seed germination for at least 26 days and (ii) when applied to *Arabidopsis* seedlings resulted in increased jasmonic acid- and salicylic acid-dependent defenses.

## 1. Introduction

*Trichoderma* is a cosmopolitan genus of filamentous fungi with an elevated interest to agriculture for its versatile beneficial effects on plants [1]. This genus includes more than 450 species [2] and is characterized by its high morphological uniformity and nutritional diversity. Mycoparasitism and competition as well as antibiosis throughout secondary metabolites (SM), including volatile organic compounds (VOCs) [3,4,5], are recognized as the main *Trichoderma* mechanisms to exert direct biocontrol of phytopathogenic fungi, oomycetes and nematodes [6,7,8]. In addition, VOCs released by *Trichoderma* can attract parasitoids and predators of insect pests [9,10]. *Trichoderma* also utilizes its competition ability in the rhizosphere where it modulates the microbiome composition [11]. In their colonization of the plant, on the roots [12] or as an endophyte [13,14], *Trichoderma* spp. have evolved the capacity to communicate with it and produce numerous multifaceted benefits ranging from growth promotion to priming of local and systemic defense responses against biotic and abiotic stresses [15,16,17]. *Trichoderma harzianum* is one of the species of most notable interest for agriculture and is the active matter of many commercial biocontrol products worldwide [1].

VOCs are small carbon-based chemicals with low molecular weight, polarity and boiling point, and high vapor pressure produced by all living beings, included fungi, which might have high biotechnological and ecological potential [18,19]. VOCs cover long distances spreading throughout the air and take part in intra and inter-specific communications, as those among fungi with other microorganisms and plants, without physical contact between them, which makes these compounds excellent infochemicals [20]. VOCs released by *Trichoderma* include lactones, ketones, alcohols, mono- and sesquiterpenes, esters and aldehydes, all of them with selective bioactivity [21,22]. Several investigations have demonstrated the antimicrobial properties of *Trichoderma* VOCs [23,24,25,26,27,28] as well as their role in the induction of defenses and/or growth promotion in plants [19,22,29,30,31,32,33,34]. Production of VOCs in *Trichoderma* is directly dependent not only on the species, but also on the strain, and it is also affected by growth conditions, developmental stage and the abiotic or biotic cues received from the environment [21,35,36,37,38]. Moreover, depending on conditions, VOCs from the same *Trichoderma* isolate can either stimulate plant growth or induce toxicity [32].

In previous studies, we reported that the disruption of the transcription factor *Thctf1* gene in *T. harzianum* T34 inhibited the production of two SMs derived from the VOC 6-pentyl-2*H*-pyran-2-one (6-PP) and compromised the antagonistic activity of this strain against *Fusarium oxysporum* [39,40]. Moreover, the disruption of *Thctf1* in this strain affects the expression of a transcriptional coactivator involved in the production of VOCs with antifungal activity [41]. In this study we have compared the VOC profiles emitted by *T. harzianum* T34, its *Thctf1* disruptant ΔD1-38 and its ectopic integration transformant ΔJ3-16 with the aim of understanding the role of THCTF1 in regulation of VOC emission in the strain T34 and whether these VOCs influence *Botrytis cinerea* antagonism, and growth and defense responses in *Arabidopsis* plants. Since a methyltransferase gene has been related to 6-PP production by *T. atroviride* [42], we have also explored the THCTF1 function in the expression of different methyltransferase genes in *T. harzianum* T34.

## 2. Materials and Methods

### 2.1. Fungal Strains and Growth Conditions

The *T. harzianum* wild type strain CECT 2413 (Spanish Type Culture Collection, Valencia, Spain), also referred to as strain T34, and the transformants ΔD1-38 and ΔJ3-16 derived from it [39], were used in this study. ΔD1-38 is the result of the disruption of *Thctf1* gene, which encodes a transcription factor previously related to the production of 6-PP derivatives, and ΔJ3-16 is consequence of the introduction of the *Thctf1* disruption cassette in another region than the *Thctf1* locus [39]. *Botrytis cinerea* 98 (*Bc*) [43] was used as target pathogen in the antifungal assay. All fungal strains were routinely grown in the dark on potato dextrose agar (PDA, Difco Laboratories, Detroit, MI, USA) medium, at 22 °C *Bc* and 25 °C *T. harzianum*. For long-term storage, the fungal conidia were maintained at −80 °C in a 30% glycerol solution.

To perform the phenotypic characterization of *Trichoderma* strains, a 5 mm diameter agar plug with the fungus was placed at the center of a Petri dish, with PDA or MEA (Malt Extract Agar, Difco Laboratories) medium, sealed or unsealed, and incubated at 22 or 25 °C. Four Petri dishes were considered for each *Trichoderma* strain and condition tested. Ten-day cultures were photographed, and conidia from PDA dishes incubated at 22 °C, both sealed and unsealed, were collected in sterile distilled water and quantified using a Thoma chamber. 

For *Trichoderma* gene expression studies, a 5 mm diameter agar plug with fungal mycelium was placed on a cellophane-covered PDA dish. Then, it was covered by the lid, sealed and incubated at 25 °C for five days. Mycelia from three biological replicates were independently collected for each *Trichoderma* strain, frozen in liquid nitrogen and stored at –80 °C until use for RNA extraction. 

### 2.2. Assay of Trichoderma VOCs Effects on ΔD1-38 Strain

To test the ability of VOCs released by either strain T34 or ΔJ3-16 to rescue the conidiation and phenotype in the disruptant ΔD1-38, we used a closed system (described below, Section 2.4.), consisting of a large Petri dish containing four small dishes. The disruptant ΔD1-38 was grown in two small dishes and the wild type or the ectopic integration transformant in the other two. The large dish was sealed and incubated at 25 °C. A second large dish containing two small dishes with ΔD1-38 and two small dishes with only PDA was used as control. Photographs were taken seven days after the inoculations. Three biological replicates per condition were set up.

### 2.3. VOCs Analyses

The VOCs released by *T. harzianum* T34, ΔD1-38 and ΔJ3-16 were analyzed by using a Proton Transfer Reaction-Quadrupole interface-Time-Of-Flight-Mass Spectrometry (PTR-Qi-TOF-MS) equipment (Ionicon Analytik GmbH, Innsbruck, Austria) in an air-conditioned room with a constant temperature of 25 ± 1 °C. To perform VOCs analysis of *Trichoderma* strains, a 5 mm diameter agar plug with the fungus was placed at the center of a 1 L Erlenmeyer conical glass flask equipped with a GL45 3-valve screw cap, containing 100 mL of PDA. VOCs protonation and measurements of headspace VOC profiles accumulated in the flasks were carried out as previously described [38]. Four different biological replicates for each sample were analyzed and the measurements were taken at five (half of the agar surface was colonized by the mycelium) and nine (all the agar surface was colonized by the mycelium) days post inoculation.

The PTR-Qi-TOF-MS raw data were acquired by the TOFDAQ Viewer software (Tofwerk AG, Thun, Switzerland), and the mass spectra and temporal ion signal profiles were extracted using the PTR-TOF-MS Viewer software (Ionicon Analytik version 3.3.8) with a custom modified Gaussian function fit for each peak. Data acquisition and peak quantification were expressed as normalized parts per billion by volume (ppbv). The calibration of the PTR spectra and the elimination of peaks associated with the PTR-MS ion source were performed as previously described [38]. The *m*/*z* signals were background-corrected by subtracting the signal obtained from the glass flasks containing only PDA. Most of the mass peaks were tentatively identified based on the available literature or by comparisons with genuine standards. 

### 2.4. Antifungal Assay of Trichoderma VOCs against Bc

An antifungal assay was used to evaluate the effect of *Trichoderma* VOCs on the growth of *Bc*. Four 5 cm diameter Petri dishes (small dish) containing PDA medium were placed inside a 15 cm diameter Petri dish (large dish), and two of the small dishes were inoculated with a plug took from the perimeter of an actively growing *Trichoderma* colony. Then, the large dish was covered by the lid, sealed, and incubated at 22 °C for four days. After that time, the other two small PDA dishes were inoculated with a plug took from the margin of a growing PDA colony of *Bc*. The large dish was covered again, sealed with two layers of Parafilm and incubated under the same conditions during two additional days. At this time point, six days after *Trichoderma* inoculation, two diameters of each *Bc* colony were recorded. In addition, large dishes containing only pathogen cultures were used as control. The experiment included three biological replicates and was repeated twice. The results are expressed as percentage of *Bc* growth inhibition by VOCs of each *Trichoderma* strain tested relative to *Bc* grown alone. 

### 2.5. Assays of Trichoderma VOCs and Arabidopsis

*Arabidopsis thaliana* Col-0 ecotype, referred to as *Arabidopsis*, was used in plant assays. Seeds were surface disinfected by soaking in a 70% ethanol and 1% (*v*/*v*) Triton X-100 solution for 20 min, followed by 10 min in 20% (*v*/*v)* bleach, and then rinsed four times with sterile distilled water. Afterwards, seeds were kept in sterile water for three days at 4 °C to break dormancy. 

The effect of *Trichoderma* VOCs on *Arabidopsis* seed germination was evaluated by using the same assay with 15 cm diameter Petri dishes as described above with some modifications. Briefly, each *Trichoderma* strain was cultured in two small dishes containing PDA. After four days of fungal incubation, twenty-six surface sterilized seeds of *Arabidopsis* were placed in each of the other two small dishes that contained Murashige and Skoog (MS, Duchefa Biochemie BV, Haarlem, The Netherlands) medium, supplemented with 1% sucrose and adjusted to pH 5.7. The large dish was sealed with two layers of plastic film and incubated at 22 °C. In parallel, a large dish containing small dishes filled with medium (but not inoculated with *Trichoderma*) was used as control. The *Arabidopsis* development was examined over 26 days, and photographs were taken at 30 days after the start of the experiment. Fresh weight data were recorded from a set of 50 twelve-day-old seedlings that were weighed together for each tested condition. Three biological replicates for each condition were set up and the experiment was repeated twice.

To evaluate the influence of *Trichoderma* VOCs on the development of *Arabidopsis* seedlings, an experiment was conducted as mentioned above, except that seedlings were grown on MS medium contained in two small dishes for seven days prior to inoculation of the fungal strain on the other two small dishes with PDA medium. This experiment was extended for two weeks and at that time point, photographs were taken, and plant material was collected, frozen in liquid nitrogen, lyophilized and stored at –80 °C for expression studies. Three biological replicates for each condition were considered. This experiment was repeated, and fourteen-day-old seedlings were collected to determine the fresh weight as indicated above.

### 2.6. Real-Time Quantitative PCR (qPCR)

Fungal or plant total RNA was extracted by using TRIZOL reagent (Invitrogen Life Technologies, Carlsbad, CA, USA), following the manufacturer’s instructions. Three biology replicates were used for each condition considered. Gene expression was analyzed by real-time quantitative PCR (qPCR). cDNA synthesis, PCR mixtures and amplification conditions were as previously described [16]. The expression levels of *lae1* (Triha1:85012), S-adenosyl methionine-dependent methyltransferases (Triha1:506014 and Triha1:81579), thiopurine S-methyltransferase (Triha1:5468) and prohibitin protein containing a methyltransferase domain (Triha1:492690) genes of *Trichoderma*, and *PR-1*, *VSP2* and *PDF1.5* genes of *Arabidopsis*, were analyzed. Primer sequences, both those that were previous described [44,45] or designed in this study, are listed in Appendix A. The Ct (cycle threshold) values were normalized with the values of the *Trichoderma* or *Arabidopsis actin* gene, and the relative gene expression was calculated using the 2^−ΔΔCT^ method [46].

### 2.7. Statistical Analyses

With the exception of the VOC data, IBM SPSS^®^ Statistics 27 (IBM Corp., Armonk, NY, USA) package was used for statistical analyses, through an analysis of variance (ANOVA) using Tukey’s test to identify significant differences among samples (*p* < 0.05). A two-way ANOVA was used to test for possible interactions between the main effects within the antifungal assay by a mean separation using Tukey’s test (*p* < 0.05).

The statistical analyses of VOCs were carried out by using Metaboanalyst platform (https://www.metaboanalyst.ca) [47], accessed on 7 september 2022. Data were normalized and autoscaled (mean-centred and divided by the standard deviation of each variable) prior to each analysis. Principal component analysis (PCA) was carried as an unsupervised method to highlight the underlying data structure. One-way ANOVA was performed coupled with Fisher’s LSD test to find significantly different means in multiple comparisons. ANOVA results were presented by heat map and hierarchical clustering in order to provide a more intuitive visualization of the VOC patterns, reordering the rows and columns so that rows (and columns) with similar profiles are closer to one another, with each entry displayed as a colour related to its signal intensity. Moreover, dendrograms were created, using Pearson correlation-based distances and the Ward’s method of agglomeration.

## 3. Results

### 3.1. Thctf1 Disruption Effect on T. harzianum Phenotype 

To verify the major phenotypic effects of *Thctf1* disruption on *T. harzianum*, the wild-type T34, ΔD1-38 and ΔJ3-16 strains were assayed for growth and conidiation on PDA and MEA media at 22 and 25 °C, using sealed and unsealed dishes. In qualitative terms, ΔD1-38 disruptant showed less aerial mycelium production than T34 and ΔJ3-16 in all assayed conditions, it being more evident on sealed dishes (Figure 1A). In addition, the loss of *Thctf1* function in *T. harzianum* affected the degree of conidiation, albeit the effect differed between sealed and unsealed conditions. When strains were cultured on unsealed PDA dishes at 22 °C, conidiation was reduced by approximately 25% in the disruptant ΔD1-38; however, the conidiation on sealed dishes at this temperature was reduced to almost zero (Figure 1B). Similar results among wild type and transformants were obtained at 28 °C (data not shown), indicating that the ΔD1-38 differential phenotype is not affected within this temperature range. A two-way ANOVA showed the effect of the variable “strain”, the variable “dish sealing”, and their combination for *Trichoderma* conidiation (*p* < 0.05).

We tested if VOCs released by *T. harzianum* strains would rescue conidiation in ΔD1-38. Growing this strain for seven days on dishes adjacent to those with T34 or ΔJ3-16, did not bring to conidia production (Appendix A), indicating that VOCs from wild type or ectopic integration strains cannot restore the phenotype of this disruptant transformant.

### 3.2. T. harzianum Thctf1 Function Loss Modifies the Expression of Methyltransferase-Related Genes

As methyltransferase LAE1 was previously related to conidiation in *T. atroviride* [42] and in view of the phenotype observed in the disruptant ΔD1-38, we have explored the role of THCTF1 in the expression of the *lae1* orthologous gene and other methyltransferase genes in *T. harzianum*. We identified in the *T. harzianum* genome an orthologous gene of *lae1* (Triha1:85012) and analyzed the expression of this gene by qPCR with the aim of knowing whether THCTF1 is involved in the production of LAE1. The *lae1* transcript was strongly down-regulated in the disruptant ΔD1-38 with respect to those of T34 or ΔJ3-16 after growing these strains in sealed PDA dishes for five days (Figure 2). 

The expression of four other genes, annotated as putative methyltransferases in the genome of *T. harzianum* (two S-adenosyl methionine (SAM)-dependent methyltransferase (Triha1:506014 and Triha1:81579) and one thiopurine S-methyltransferase (Triha1:5468)), or harboring a methyltransferase domain (prohibitin protein (Triha1:492690)), was also analyzed by qPCR. Results showed that, compared to T34 or ΔJ3-16, the expression levels of three out of these four genes were significantly lower in the disruptant (Figure 2), indicating that THCTF1 is involved in the expression of methyltransferase-related genes in *T. harzianum.*

### 3.3. Differences in T. harzianum VOC Profiles

We compared the VOC production among wild type T34, ectopic integration ΔJ3-16 and disruptant ΔD1-38 strains grown on PDA medium five and nine days after the inoculation. A total of 129 and 130 VOCs were detected at five and nine days, respectively, in the range of measured masses (mass protonated range *m*/*z* = 20–300) after the subtraction of peaks associated with the PTR-MS ion source and their isotopes. VOCs’ putative identification is reported in Appendix A [48,49,50,51,52,53,54,55,56,57,58,59,60,61,62,63,64,65,66,67,68,69,70,71]. Overall, all samples’ released VOCs were blends composed of similar compounds, but in different proportions. The tentatively identified compounds belong to different chemical classes such as alcohols, aldehydes, esters, organic acids, ketones and sulfur compounds. 

Throughout the experiment, VOC emissions of the three *T. harzianum* strains varied (Figure 3 and Figure 4). Even if PCA analyses of the two collection times gave slightly different results there is a general trend, as VOC profiles emitted by T34 and ΔJ3-16 were partially overlapping both at five (with the first three principal components explaining the 67.2% of total variance, Figure 3A) and nine days (with the first three principal components explaining the 77.4% of total variance, Figure 3B). This indicates that samples from these two genotypes share some characteristics, while the ΔD1-38 disruptant resulted clearly separated, showing an altered production of VOCs.

A detailed analysis of the compounds at five and nine days reveals that those common to all three *T. harzianum* strains such as ethanol and acetaldehyde did not vary during the experiment and accounted for more than 90% of total VOCs.

The one-way ANOVA and post hoc test on five- and nine-days data (Appendix A) showed significant differences (*p* < 0.05) in 43 and 53 out of 129 and 130 detected peaks, respectively. The peak *m*/*z* 93.068 was the only one not shared among the two time point data sets: it was detected only at 9 days. Hierarchical heat map clusters of ANOVA significant VOCs are presented in Figure 4. The heat maps for both time points show well-defined clusters of volatiles characteristically over- or under-emitted among the three fungal strains. At five days (Figure 4A), the disruptant ΔD1-38 over-emitted a cluster of volatiles consisting mostly of VOCs that showed significant differences (*p* < 0.05) with those released by both T34 and ΔJ3-16 strains (*m*/*z* 61.026, *m*/*z* 129.090, *m*/*z* 79.038, *m*/*z* 101.059, *m*/*z* 147.138, *m*/*z* 129.127, *m*/*z* 87.080, *m*/*z* 119.049, *m*/*z* 43.012, *m*/*z* 40.026, *m*/*z* 265.856, *m*/*z* 77.059, *m*/*z* 107.068, *m*/*z* 145.122, *m*/*z* 159.138, *m*/*z* 54.033, *m*/*z* 39.024, *m*/*z* 59.051, *m*/*z* 85.064, *m*/*z* 127.112, *m*/*z* 266.856). At nine days (Figure 4B), the disruptant over-emitted most of the 53 VOCs significantly different among the three strains. Overall, the disruptant emitted much more VOCs in comparison to wild type and ectopic integration strains.

### 3.4. THCTF1 Affects the Production of Fungistatic VOCs of T. harzianum

The effect of VOCs released by *T. harzianum* T34, ΔJ316 and ΔD1-38 strains on the growth of the phytopathogen *B. cinerea (Bc)* was evaluated in sealed PDA dishes (Appendix A). The *Bc* colony growth was significantly reduced by T34 and ΔJ3-16 VOCs compared with that obtained in the control without *Trichoderma* (Figure 5), inhibition percentages being 26.72 ± 8.38 and 20.34% ± 4.19, respectively. Nevertheless, no growth differences were observed when the pathogen was exposed to VOCs released by the disruptant strain compared to the control.

### 3.5. Effects of Trichoderma VOCs on Arabidopsis Seed Germination, Seedling Development and Defense Gene Expression

The effect of VOCs emitted by *T. harzianum* on *Arabidopsis* seed germination and seedling development was assessed on co-culture experiments in closed MS plus 1% sucrose dishes. After 7 days, the germination of the control seeds was 97% and none of the treatments showed germination. At 26 days, seeds exposed to T34 or ΔJ3-16 VOCs reached germination rates (97%) similar to control, while seeds exposed to ΔD1-38 VOCs were unable to germinate (Figure 6), indicating that volatiles of the disruptant strain may have toxic effects on *Arabidopsis* seeds. As would be expected due to the delay in germination, fresh weight values of 12 day old seedlings subjected to the VOCs of T34 or ΔJ3-16 were lower than those of the control (Table 1).

In a different experiment, seven-day-old *Arabidopsis* seedlings were exposed for seven days to *T. harzianum* VOCs to determine their effects on plant growth (Figure 7). Seedlings subjected to VOCs of T34 or ΔJ3-16 showed increased development in terms of more vigor, greenery and size than those of the control, although no significant differences were detected in their fresh weight measurements (Table 1). In the case of seedlings exposed to VOCs of ΔD1-38, a phenotype with smaller size and yellowing was observed (Figure 7) and their fresh weight values were much lower than those of the rest of the seedlings in the assay (Table 1). These results could indicate that a functional *Thctf1* gene is necessary to *Trichoderma* VOC production that do not have negative effects on plant growth. 

The expression of genes related to plant immunity was also analyzed in seedlings exposed to *T. harzianum* VOCs for seven days (Figure 8). Seedlings subjected to ΔD1-38 VOCs showed the highest expression levels of marker genes for salicylic acid (SA)- (*PR-1*), jasmonic acid (JA)- (*VSP2*) and JA/ethylene (ET)- (*PDF1.5*) dependent defense pathways. No differences were observed in the expression levels of these three genes between control plants and those exposed to T34 VOCs and between plants subjected to the VOCs of T34 and ΔJ3-16.

## 4. Discussion

In a previous study, we found that the transcription factor THCTF1 of *T. harzianum* was found to be involved in the production of two 6-PP derivatives and the antagonism of this fungus against *Fusarium oxysporum* [39]. The unsaturated lactone 6-PP is responsible for the characteristic “coconut aroma” of certain *Trichoderma* species and it has been related to the fungistatic activity of *Trichoderma* strains [29,36]. Since 6-PP and its analogs are VOCs, in the present study we have analyzed the volatilome emitted by three *T. harzianum* strains: T34 (wild type), ΔD1-38 (*ΔThctf1* disruptant) and ΔJ3-16 (ectopic integration transformant), on PDA at two different growth times, five and nine days. As might be expected, the PTR-Qi-TOF-MS analysis showed that many VOCs were present in similar abundance in samples from the three *T. harzianum* strains, not showing significant differences at the two time points considered, such as those putatively identified as decanal, acetaldehyde, 2,4-dimethylfuran, furan, formaldehyde, 1,2,4-trimethylbenzene, 6,10-dimethyl-5,9-undecadien-2-one (geranylacetone), *p*-cymene, methanethiol, 3-methylacetophenone, 2-methoxy-4-methylphenol (=creosol), sesquiterpenes, ethanol, methanol, hexenal isomeres and cedrol, molecules all known to be normally emitted by different microbes, including *Trichoderma* spp. [18,38,50,72]. However, no significant differences in the production levels of 6-PP and its derivatives were detected among the three strains of *T. harzianum* under the experimental condition of the present study. This would not be surprising as it has been described that microbial VOC emissions are dependent on substrate composition, biotic environment, growth phase and life cycle [36,73].

We found that ΔD1-38 disruptant showed less aerial mycelium and a reduced conidiation in comparison to both the wild type and the ectopic integration transformant. This fact could be due to how gene disruption affects the regulation of master genes, as occurs with the downregulation of *lae1*, evidenced by the lower concentration of conidia detected on unsealed plates, without accumulation of VOCs. Moreover, gene disruption modifies the production of VOCs which, when accumulating in the sealed plates, may affect genes involved in conidiation. In fact, the formation of conidia by the disruptant was reduced to almost zero in a closed system in which the VOCs could not spread freely. These phenotypic differences did not depend on temperature or culture medium composition. It has been described that *Trichoderma* conidiation may be triggered by VOCs from adjacent *Trichoderma* colonies [74]. However, the VOCs from T34 or ΔJ3-16 were unable to rescue conidiation in ΔD1-38 disruptant. These authors suggested that the production of 1-octen-3-ol, 3-octanol and 3-octanone is linked to the process of conidia formation. We have found that 1-octen-3-ol/3-octanone (*m*/*z* 129.127) was over-emitted in the disruptant compared to T34 and ΔJ3-16 strains. However, ranking the emitted VOCs as percentages of the total (data not shown), this compound has a peak of emission in the wild type going from position 102nd at five days (starting conidiation phase) to position 18th at nine days, while in the disruptant it is kept almost in the same position in the ranking, around 40th. These results suggest that for the conidiation process not only is the presence of certain VOCs important, but also their relative concentration in the total blend of volatiles. 

It has been reported that pyrone derivatives are major components of the volatilome of *T. atroviride* [75] and that the disruption of *lae1* gene, which encodes a methyltransferase, almost entirely reduced the conidiation of this fungus and, in agreement with our results, it could not be rescued by VOCs from the parent strain [42]. Since it has also been described that *lae1* is related to 6-PP production and to the *T. atroviride* antagonism [42], we searched for a *lae1* orthologue in the publicly available genome of *T. harzianum* CBS 226.95 [76]. We compared the expression of the *lae1* orthologue in the three *T. harzianum* strains and found that it was lower in the disruptant strain ΔD1-38. Gene expression results obtained by qPCR for the three strains allow us to relate THCTF1 function with some but not all methyltransferase-related genes present in the genome of *T. harzianum*. Moreover, *lae1* gene deletion in *T. atroviride* resulted also in a reduction in the expression of polyketide synthase (PKS)-encoding genes [42]. These enzymes have a fundamental role in the production of non-ribosomal peptides (NRPs) or polyketides, molecules known to be involved in antimicrobial activity of some fungi [77]. The 4-phosphopantetheinyl transferase of *T. virens* plays a role in plant protection against *Bc* through VOC emission. The *lae1* gene looks to be a master regulator that is quite conserved and expanded in the genus *Trichoderma,* but the function of the other potential methyltransferases cannot be ruled out and therefore might need to be the target for future experiments.

The ability of *Trichoderma* VOCs to inhibit microbial growth has been documented with respect to different plant pathogenic fungi. For instance, VOCs produced by *T. harzianum* T-E5 caused a remarkable decline in *F. oxysporum* f. sp. *cucumerinum* growth [23], and those from *T. asperelloides* TSU1 inhibited the growth of different ascomycetes [28]. A delay in conidial germination and a suppression of germ tube elongation of *Bc* and *F. oxysporum* has also been related to the VOCs emitted by *T. koningiopsis* T-51 [20]. Our results have shown that VOCs from T34 or ΔJ3-16 were able to inhibit the growth of *Bc* while those of the disruptant ΔD1-38 did not. Although pyrones are the most studied *Trichoderma* volatiles, the antifungal capacity of *Trichoderma* VOCs is not unique to 6-PP and its derivatives [23]. Thus, in addition to 6-PP, 2-methyl-1-butanol and 2-pentyl furan have been suggested as VOCs responsible for growth inhibition of fungi such as *Colletotrichum*, *Sclerotium* or *Penicillium* by *T. asperelloides*, 2-pentyl furan being the most abundant of the antifungal blend [34]. The VOCs 2-pentyl furan (*m*/*z* 139.112) and benzaldehyde (*m*/*z* 97.049) were over-emitted in T34 and ΔJ3-16 blends from axenic cultures in comparison with those of ΔD1-38 disruptant, which would support that these VOCs are related with the antifungal activity of *T. harzianum* against *Bc*. Moreover, VOC profiles of *Trichoderma* spp. are susceptible to change in interaction with potential preys [27,78,79], as occurs with selinene, limonene and cyclohexane that were identified from co-cultivation scenarios of *T. harzianum* and *Laccaria bicolor* compared to the respective axenic cultures [79]. Nevertheless, the results observed with *Bc* cannot be extrapolated to other fungal pathogens, and a more extensive and case-by-case study should be conducted once the antagonistic potential of the VOCs regulated by THCTF1 has been proven.

*Trichoderma* VOCs are able to promote plant growth and immunity [20,29,30,34], although the outcome depends on its concentration [29,68,80]. The biostimulant capacity of VOCs released by *Trichoderma* has been accompanied by increased levels of endogenous sugars in shoots, roots and root exudates of *Arabidopsis* plants [22]. Conversely, it has been reported that there is a decrease in fresh shoot weight and chlorophyll in *Arabidopsis* plants exposed to VOCs of *T. atroviride* CBS 01.209 [33]. According to this work, we observed that *T. harzianum* VOCs delayed germination of *Arabidopsis* seeds. Moreover, ΔD1-38 VOCs arrested seed germination for at least the 26 days the experiment lasted. We cannot attribute this inhibition of seed germination by the disruptant to a specific VOC, since it is known that VOCs work synergistically to exert their activity [30], but some of the volatiles reported to have an inhibitory effect on *Arabidopsis* seed germination were over-emitted in ΔD1-38 disruptant compared to the other two strains. Indeed, octanal/1-octen-3-ol/3-octanone (*m*/*z* 129.127), 2-octenal/1-octen-3-one (*m*/*z* 127.112) and pentanal/pentanone (*m*/*z* 87.080), likely affecting *Arabidopsis* seeds germination [37], were differentially over-emitted in the disruptant at five and nine days. As previously reported [32,33], we have observed that VOCs of *T. harzianum* T34 exert a positive effect on *Arabidopsis* seedlings and their beneficial effects on plants depends on the developmental stage. The deleterious phenotype observed in plants subjected to ΔD1-38 VOCs could be demonstrative of the important role of *Thctf1* gene in the production of VOCs with positive effects on *Trichoderma*-plant interaction. 

It has been described that *Trichoderma* VOCs can activate defense responses in plants and, although little is known about the overall plant gene expression in response to VOCs emitted by this fungus, the role of some individual *Trichoderma* VOCs has been reported. Thus, the exposure to 6-PP induced the expression of the SA-responsive marker *PR-1* in *Arabidopsis* [29], while 1-octen-3-ol enhanced the expression of ET- and JA-dependent defense genes [81]. These two *Trichoderma* VOCs can attract parasitoids and predators of insect pests and confer antifeedant effects on plants that mitigate herbivore attack [10,82]. By contrast, *Arabidopsis* plants exposed to 1-decene showed a reduced expression of the JA biosynthesis-related *LOX4* gene accompanied by the downregulation of WRKY transcription factors involved in defense [37]. We have observed in *Arabidopsis* plants exposed to a VOC blend of ΔD1-38 disruptant an upregulation of the three defense marker genes analyzed. This is indicative that the plant perceives the ΔD1-38 VOC blend as a stress signal, which in addition to limiting plant development, it activates both SA- and JA-dependent defenses. It has been reported that exogenous application of the *Bacillus subtilis*-derived elicitor acetoin (3-hydroxy-2-butanone) was able to trigger systemic resistance in *Arabidopsis*, with an increase in the expression of the defense related genes *PDF1.2* and *PR-1* [83], which suggests the activation of SA/ET pathways. We have observed that ΔD1-38 disruptant over-emitted acetoin (*m*/*z* 89.59) and overexpressed *PDF1.5* and *PR-1* genes, in comparison to wild type and ectopic transformant strains, confirming that this volatile has a role in the activation of plant defense responses.

It has been reported in *Arabidopsis* that VOCs of *T. virens* increase the expression of a JA-defense marker gene without changing the expression of another, in theory antagonistic, SA marker gene [77], but it has also been shown that VOCs of *T. asperellum* raise the levels of SA and abscisic acid marker genes whereas those of JA remained unaltered [31]. We have seen that plants subjected to VOCs of *T. harzianum* T34 or ΔJ3-16 have a larger size with no change in the expression of immunity marker genes for SA-, JA- or ET-dependent defenses, compared to control plants. These observations are illustrative of the difficulty of interpreting the variation of plant growth and defense responses as a result of the different stimuli received by distinct *Trichoderma* strains.

## 5. Conclusions

The activity of a transcription factor may affect different functions involved in a variety of biological processes. In our case, the use of a *Thctf1* null transformant has made it possible to link the transcription factor THCTF1 with conidiation, expression of methyltransferase-related genes and VOC production in *T. harzianum.* VOCs affected by THCTF1 are involved in the *T. harzianum* antifungal capacity and have effects on plant defense and development. As VOCs are emitted as blends, it is difficult to know whether a particular outcome is caused by a single VOC, especially when the observed effects are the result of multifaceted interactions. Moreover, there are limitations associated with the analytical methods used for VOC identification and the VOCs responsible of an effect may be undetected by the selected analytical method. Due to the potential of beneficial fungi such as *Trichoderma* in agriculture, more thorough studies are needed to decipher the role of VOCs in the context of their interaction with plants and other organisms in natural systems.

## Figures and Tables

**Figure 1 jof-09-00654-f001:**
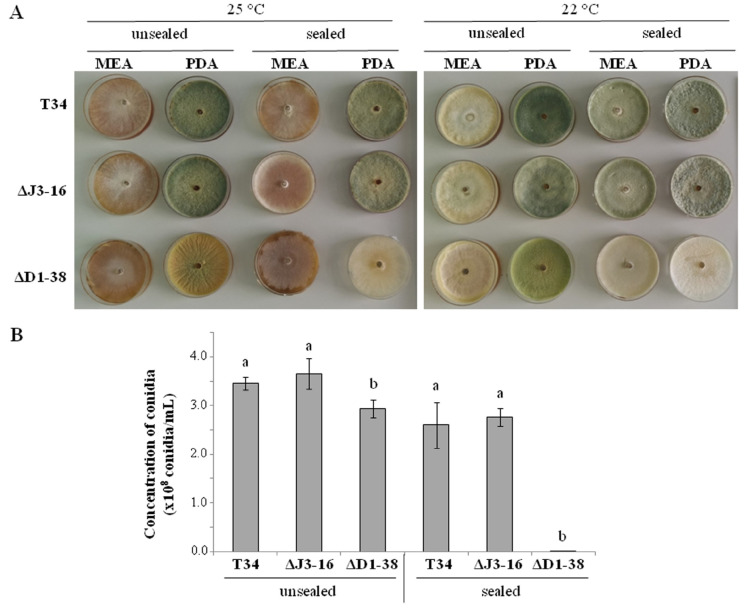
Phenotype and conidiation of *Trichoderma harzianum* strains (wild type T34, ectopic integration transformant ΔJ3-16 and *Thctf1* disruptant ΔD1-38). (**A**) Phenotype of T34, ΔJ3-16 and ΔD1-38 after growing on MEA or PDA medium, unsealed or sealed dishes, at 25 (left) or 22 °C (right) for ten days. (**B**) Quantification of conidiation of T34, ΔJ3-16 and ΔD1-38 on PDA incubated at 22 °C for ten days, unsealed (left) and sealed (right) dishes. Values are means of four biological replicates. For each growth condition (unsealed or sealed), different letters above indicate significant differences according to one-way analysis of variance (ANOVA) followed by Tukey’s test (*p* < 0.05). Significant effects were determined by a two-way ANOVA for *T. harzianum* strain, dish sealing, and their combination (*p* < 0.05).

**Figure 2 jof-09-00654-f002:**
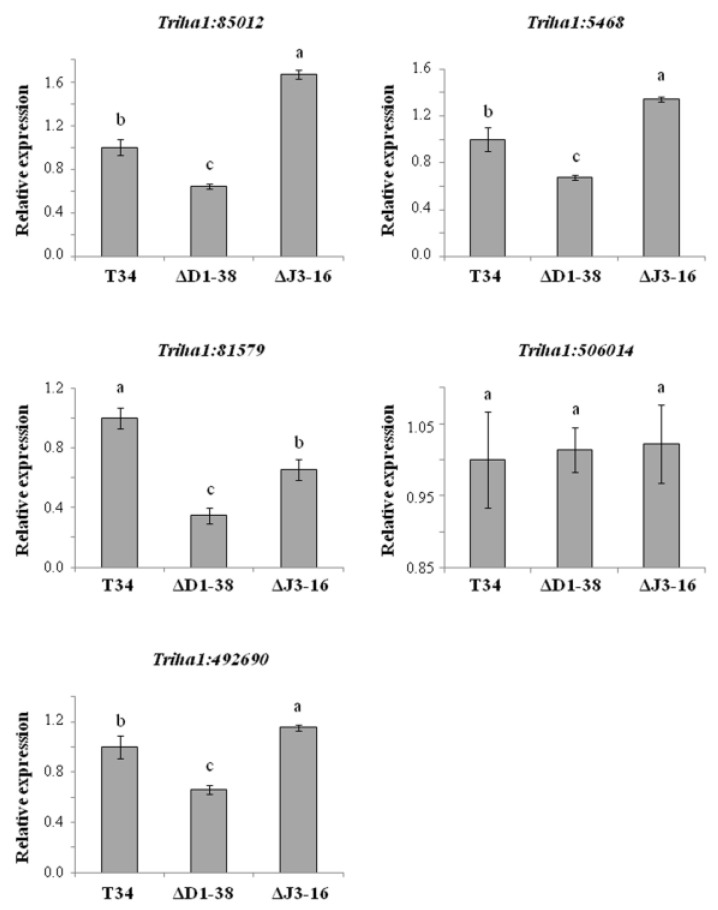
Relative expression of methyltransferase-related genes in *Trichoderma harzianum* strains (wild type T34, *Thctf1* disruptant ΔD1-38 and ectopic integration transformant ΔJ3-16) by real-time quantitative PCR. Strains were cultured on PDA sealed dishes for five days. Analyzed genes: *lae1* (Triha1:85012), two S-adenosyl methionine-dependent methyltransferases (Triha1:506014 and Triha1:81579), one thiopurine S-methyltransferase (Triha1:5468) and a prohibitin protein containing a methyltransferase domain (Triha1:492690). *T. harzianum* T34 *actin* was used as endogenous gene. For each gene, values correspond to relative measurements against the transcript in T34 (2^−ΔΔCT^ = 1) and data are the mean values for three biological replicates. Different letters above the bars indicate significant differences according to one-way analysis of variance (ANOVA) followed by Tukey’s test (*p* < 0.05).

**Figure 3 jof-09-00654-f003:**
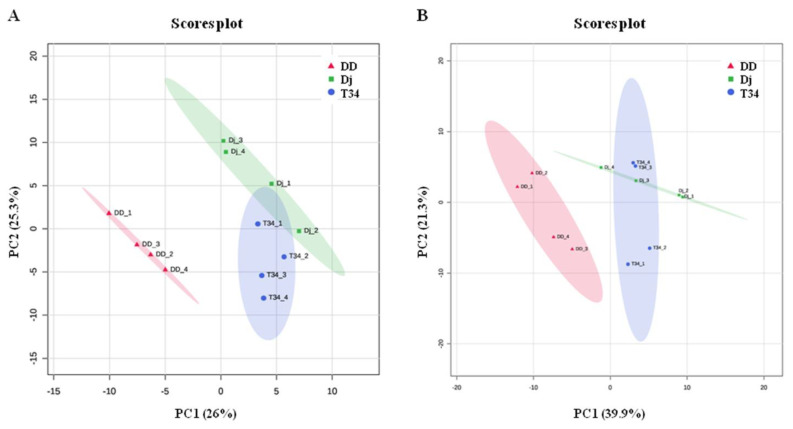
First two components of the PCA analysis of the PTR-Qi-TOF-MS data of VOCs emitted by three *Trichoderma harzianum* strains (T34 = wild type; Dj = ΔJ3-16 ectopic integration transformant; DD = ΔD1-38 disruptant transformant) on PDA culture medium. (**A**) Five and (**B**) nine days post inoculation. The variance explained by each component is reported in brackets.

**Figure 4 jof-09-00654-f004:**
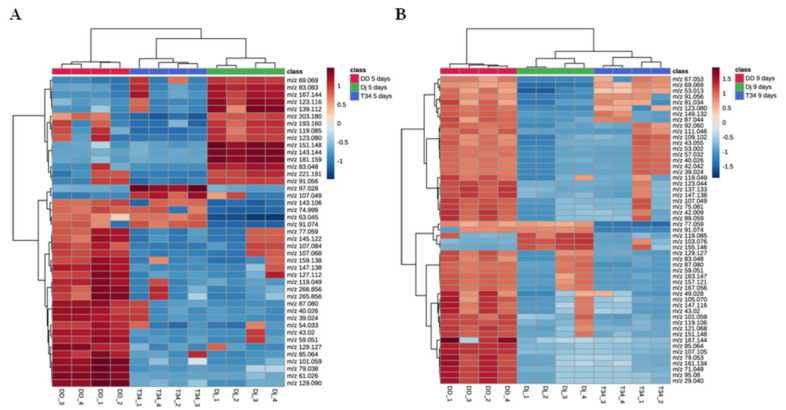
Heat maps and two-dimensional hierarchical dendrograms of VOCs emitted by *Trichoderma harzianum* strains (T34 = wild type, Dj = ΔJ3-16 ectopic integration transformant; DD = ΔD1-38 disruptant transformant) on PDA culture medium. (**A**) Five and (**B**) nine days. Biological replicates are in columns and variables are in rows. Each colored cell on the map corresponds to a concentration value, following a blue/red chromatic scale from −2 value (very low production) to 2 (extremely high production). Only analysis of variance significant peaks was used (*p* < 0.05). Pearson distance and Ward’s clustering algorithm were used for dendrograms. The tentative identification and chemical group of VOCs are detailed in Appendix A.

**Figure 5 jof-09-00654-f005:**
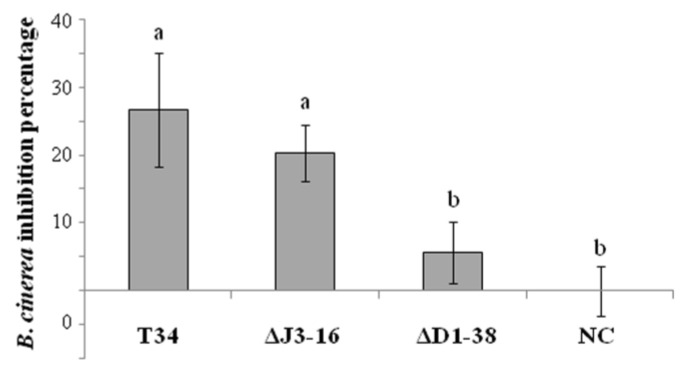
Inhibition percentages of colony diameter of *Botrytis cinerea* (*Bc*) exposed to VOCs from *Trichoderma harzianum* strains (wild type T34, ectopic integration transformant ΔJ3-16 and *Thctf1* disruptant ΔD1-38) for two days. *Bc* cultures unexposed to *T. harzianum* VOCs were used as control (NC). Values are means of three biological replicates, and different letters above the bars indicate significant differences according to one-way analysis of variance (ANOVA) followed by Tukey’s test (*p* < 0.05).

**Figure 6 jof-09-00654-f006:**
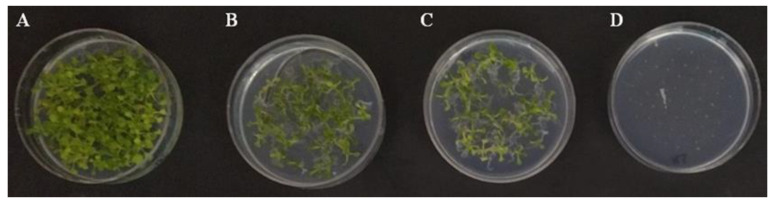
Germination of *Arabidopsis thaliana* seeds in MS medium after 26 day exposure to VOCs from *Trichoderma harzianum* PDA cultures. (**A**) Control, seeds exposed to uncultured PDA medium, and exposed to VOCs released by PDA cultures from wild type T34 (**B**), ectopic integration transformant ΔJ3-16 (**C**) and *Thctf1* disruptant ΔD1-38 (**D**).

**Figure 7 jof-09-00654-f007:**
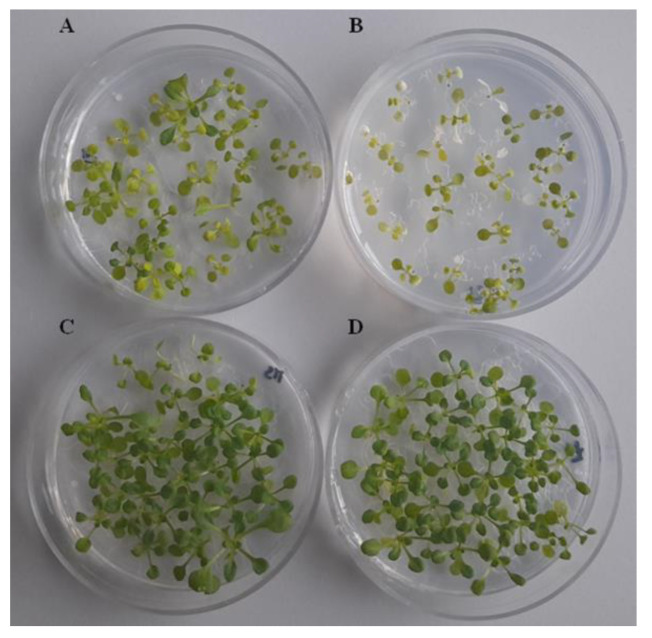
*Arabidopsis thaliana* 14 day seedling phenotypes developed in MS medium after sharing atmosphere with *Trichoderma harzianum* VOCs for seven days. (**A**) Control, seedlings exposed to uncultured PDA medium, and exposed to VOCs released by PDA cultures from *Thctf1* disruptant ΔD1-38 (**B**), wild type T34 (**C**) and ectopic integration transformant ΔJ3-16 (**D**).

**Figure 8 jof-09-00654-f008:**
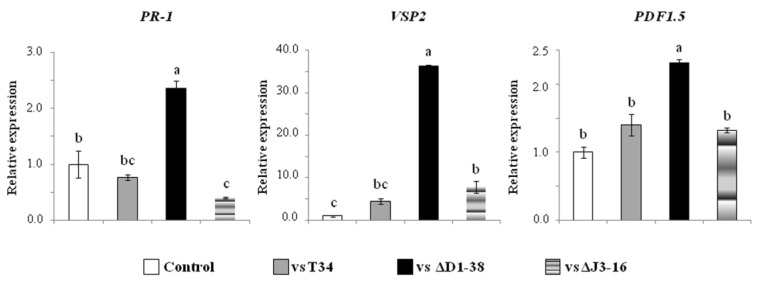
Relative expression of defense-related genes in 14 day *Arabidopsis* seedlings exposed for seven days to VOCs from *Trichoderma harzianum* (wild type T34, *Thctf1* disruptant ΔD1-38 or ectopic integration transformant ΔJ3-16) PDA cultures. Analyzed genes: *PR-1* (pathogenesis-related protein 1), *VSP2* (vegetative storage protein 2) and *PDF1.5* (plant defensin 1.5). *Arabidopsis actin* was used as endogenous gene. For each gene, values were referred to unexposed plants used as control (2^−ΔΔCT^ = 1). Data are the mean values for three biological replicates. Different letters above the bars indicate significant differences according to one-way analysis of variance (ANOVA) followed by Tukey’s test (*p* < 0.05).

**Table 1 jof-09-00654-t001:** Average fresh weight (mg/seedling) of three biological replicates of *Arabidopsis* seedlings (n = 50) exposed to *Trichoderma* VOCs (wild type T34, ectopic integration transformant ΔJ3-16 and *Thctf1* disruptant ΔD1-38). (A) *Arabidopsis* seeds were sown in MS medium and directly exposed to *Trichoderma* VOCs. Fresh weight values were taken twelve days after VOC exposition. (B) *Arabidopsis* seeds were sown in MS medium and, seven days after sowing, seedlings were exposed to *Trichoderma* VOCs. Fresh weight values were taken seven days after VOC exposition. Controls were grown without *Trichoderma* VOC exposition. For each experiment (A or B) different letters above indicate significant differences according to one-way analysis of variance (ANOVA) followed by Tukey’s test (*p* < 0.05).

	Fresh Weight (mg/Seedling)	
*Trichoderma* VOCs	A	B
Control	4.3 ± 0.7 ^a^	7.3 ± 2.1 ^a^
T34	2.5 ± 0.7 ^b^	8.9 ± 0.1 ^a^
ΔJ3-16	2.5 ± 0.1 ^b^	8.9 ± 1.8 ^a^
ΔD1-38	ND ^c^	4.2 ± 0.7 ^b^

ND: Not detected fresh weight since seeds did not germinate.

## Data Availability

Not applicable.

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
