# Peer review of "Trichoderma harzianum Volatile Organic Compounds Regulated by the THCTF1 Transcription Factor Are Involved in Antifungal Activity and Beneficial Plant Responses"

_jof, 2023, doi:10.3390/jof9060654_

Round 1
Reviewer 1 Report
In the manuscript entitled Trichoderma harzianum volatile organic compounds regulated by the THCTF1 transcription factor are involved in antifungal activity and beneficial plant responses, M. Belén Rubio and colleagues study a transcription factor THCTF1 from Trichoderma harzianum. Thctf1 disruption inhibited conidiation and affected the production of many VOCs. VOCs released by T. harzianum ΔD1-38 strains affect the growth of the phytopathogen Botrytis cinerea (Bc), seed germination and seedling development in Arabidopsis.
Overall, the experiments are well-designed, and the data are convincing. Here are some suggestions for the authors to improve their study.
Line 208: Do the authors have any ideas as to why the conidiation of the disruptant ΔD1-38 on sealed dishes at 22 °C was reduced almost zero (There was no significant difference in the number of conidia produced by either wild-type T34 or the ectopic holotype transformant ΔJ3-16 on sealed and unsealed dishes)?
Line 299: What is the logic behind examining THCTF1 function in the expression of different methyltransferase genes in T. harzianum T34?
Line 310: Why was B. cinerea (Bc) chosen over other pathogens in antifungal assay of Trichoderma VOCs? It is also not entirely clear why the authors did not use multiple pathogens for this assay.
Line 325: Why did the authors not show the germination rate of Arabidopsis seeds under the effects of VOCs? In addition, I would prefer that authors to show the growth and development indicators of Arabidopsis seedlings, such as plant height, fresh weight, etc.
Line 355: This assay of VOCs induced defense response in Arabidopsis would be more convincing if the authors were able to figure out the response of Arabidopsis to pathogens under VOC treatment.
Author Response
Referee 1:
Thank you very much for your positive comments. These are our answers to the questions asked point by point:
1.- Line 208: Do the authors have any ideas as to why the conidiation of the disruptant ΔD1-38 on sealed dishes at 22 °C was reduced almost zero (There was no significant difference in the number of conidia produced by either wild-type T34 or the ectopic holotype transformant ΔJ3-16 on sealed and unsealed dishes)?
Our explanation is that there could be two reasons: i) Gene disruption may affect the regulation of master genes of sporulation, as occurs with the downregulation of lae1, evidenced by the lower concentration of conidia detected on unsealed plates, without accumulation of VOCs; ii) The near absence of conidiation on the sealed plates may be due to the cumulative effect, compared to the previous possibility, of the mixture of differentially accumulated VOCs in the sealed disruptant plateS that would be affecting genes involved in conidiation.
We have included the following new text (New Lines 414-419): “This fact could be due to gene disruption affects the regulation of master genes, as occurs with the downregulation of lae1, evidenced by the lower concentration of conidia detected on unsealed plates, without accumulation of VOCs. Moreover, gene disruption modifies the production of VOCs which, when accumulating in the sealed plates, may affect genes involved in conidiation”.
2.- Line 299: What is the logic behind examining THCTF1 function in the expression of different methyltransferase genes in T. harzianum T34?
We think the reviewer is referring L229. As methyltransferase LAE1 was previously related to conidiation in T. atroviride (Aghcheh et al., 2013), and in view of the phenotype observed in the disruptant T. harzianum ΔD1-38, we have explored the role of THCTF1 in the expression of the lae1 orthologous gene and other methyltransferase genes in T. harzianum.
We have modified the text in New Lines 234-238: “As methyltransferase LAE1 was previously related to conidiation in T. atroviride [42] and in view of the phenotype observed in the disruptant ΔD1-38, we have explored the role of THCTF1 in the expression of the lae1 orthologous gene and other methyltransferase genes in T. harzianum. We identified in the T. harzianum genome an orthologous gene of lae1 (Triha1:85012), and analyzed the…”
3.- Line 310: Why was B. cinerea (Bc) chosen over other pathogens in antifungal assay of Trichoderma VOCs? It is also not entirely clear why the authors did not use multiple pathogens for this assay.
Our aim has been to study whether disruption of the Thctf1 gene affects the production of volatiles with antifungal activity. As in a previous study (Rubio et al., 2009) several antagonism tests were performed with the wild strain and the disruptant strain against three different pathogens, although no volatile tests were included, we now want to explore whether these molecules play a role in the biocontrol exerted by T. harzianum. Since B. cinerea was the pathogen on which the tested Trichoderma strains showed the least activity (Rubio et al., 2009), we thought it would be interesting to evaluate the biocontrol potential via VOCs against this pathogen. In any case, the results observed with B. cinerea cannot be extrapolated to other fungal pathogens and a more extensive study can be done once it has been demonstrated that VOCs produced under the control of THCTF1 are involved in biocontrol by T. harzianum.
We have included this text in New Lines 469-472: “Nevertheless the results observed with Bc cannot be extrapolated to other fungal pathogens and a more extensive and case-by-case study should be done once the antagonistic potential of the VOCs regulated by THCTF1 has been proven”.
4.- Line 325: Why did the authors not show the germination rate of Arabidopsis seeds under the effects of VOCs? In addition, I would prefer that authors to show the growth and development indicators of Arabidopsis seedlings, such as plant height, fresh weight, etc.
The reviewer is right as this information can be made available. We have produced a new Table 1 to show fresh weight data of Arabidopsis plants after exposure to VOCs from the different Trichoderma strains in two types of assays, on seeds and seedlings. Data on seed germination rates have been included in the text.
This is the new text:
- New Lines 159-160: “Fresh weight data were recorded from a set of 50 twelve-days-old seedlings that were weighed together for each tested condition”.
- New Lines 168-170: “This experiment was repeated and fourteen-days-old seedlings were collected to determine the fresh weight as indicated above”.
- New Lines 332-338: “After 7 days, the germination of the control seeds was 97% and none of the treatments showed germination. At 26 days, seeds exposed to T34 or ΔJ3-16 VOCs reached germination rates (97%) similar to control, while seeds exposed to ΔD1-38 VOCs were unable to germinate (Figure 6), indicating that volatiles of the disruptant strain may have toxic effects on Arabidopsis As would be expected due to the delay in germination, fresh weight values of 12-day-old seedlings subjected to the VOCs of T34 or ΔJ3-16 were lower than those of the control (Table 1)”.
- The legend of Table 1 is (New Lines 346-354): “Average fresh weight (mg/seedling) of three biological replicates of Arabidopsis seedlings (n=50) exposed to Trichoderma VOCs (wild type T34, ectopic integration transformant ΔJ3-16 and Thctf1 disruptant ΔD1-38). (A) Arabidopsis seeds were sown in MS medium and directly exposed to Trichoderma Fresh weight values were taken twelve days after VOC exposition. (B) Arabidopsis seeds were sown in MS medium and, seven days after sowing, seedlings were exposed to Trichoderma VOCs. Fresh weight values were taken seven days after VOC exposition. Controls were grown without Trichoderma VOC exposition. For each experiment (A or B) different letters above indicate significant differences according to one-way analysis of variance (ANOVA) followed by Tukey’s test (p < 0.05)”.
- New Lines 360-361): “…although no significant differences were detected in their fresh weight measurements (Table 1)”.
- Lines 346-347 (New Lines 350-353): “…and their fresh weight values were much lower than those of the rest of the seedlings in the assay (Table 1). These results could indicate that a functional Thctf1 gene is necessary to Trichoderma VOC production that do not have negative effects on plant growth”.
- New Line 488: “… a positive effect on Arabidopsis seedlings and…”
5.- Line 355: This assay of VOCs induced defense response in Arabidopsis would be more convincing if the authors were able to figure out the response of Arabidopsis to pathogens under VOC treatment.
Surely it would indeed be more convincing. We have not done the pathogen application assays because we do not have an experimental method to break the closed system to introduce the pathogen without the escape of VOCs. But it is certain that if VOCs induce systemic plant defence, new work can be designed to challenge Trichoderma VOC-primed plants with pathogens.
Reviewer 2 Report
The manuscript entitled "Trichoderma harzianum volatile organic compounds regulated by the THCTF1 transcription factor are involved in antifungal activity and beneficial plant responses" show the relationship between the THCTF1 transcription factor with conidiation and VOC production. I liked the experiments shown in the results, even though the results were not conclusive. I'm aware of how hard it is to perform such experiments. The manuscript is very well written and organized. My only two suggestions are:
- Generate better-quality images, especially for figures 3 and 4. It's impossible to read the text there.
- Are these VOCs derived from PKS/NRPS? If known for some of them, add a short piece of text mentioning this.
I have found one error in the lae1 protein id, the correct one for Triha1 is 85012. I've also noticed that the authors tested the expression of four other methyltransferases, finding similar expression patterns. What about the paralogs of lae1? This master regulator is quite conserved and expanded in Trichoderma (https://mycocosm.jgi.doe.gov/clm/run/Trilon1-comparative-qc.2640/12;Q1rSB-?organism=Trilon1). If not feasible for this publication, it could be a good target for future studies.
Author Response
Referee 2:
1.- Generate better-quality images, especially for figures 3 and 4. It's impossible to read the text there.
Done. We have produced new images with a better quality.
2.- Are these VOCs derived from PKS/NRPS? If known for some of them, add a short piece of text mentioning this.
According to reviewer’s suggestion we added a paragraph on PKs/NRPs. Unluckily we cannot argue if differentially emitted VOCs could be specifically related to PKs/NRPs.
We have included this text (New Lines 442-447): “Moreover, Lae1 gene deletion in T. atroviride resulted also in a reduction of the expression of polyketide synthase (PKS)-encoding genes [42]. These enzymes have a fundamental role in the production of non-ribosomal peptides (NRPs) or polyketides, molecules known to be involved in antimicrobial activity of some fungi [83]. The 4-phosphopantetheinyl transferase of T. virens plays a role in plant protection against Bc through VOC emission”.
3.- I have found one error in the lae1 protein id, the correct one for Triha1 is 85012. I've also noticed that the authors tested the expression of four other methyltransferases, finding similar expression patterns. What about the paralogs of lae1? This master regulator is quite conserved and expanded in Trichoderma (https://mycocosm.jgi.doe.gov/clm/run/Trilon1-comparative-qc.2640/12;Q1rSB-?organism=Trilon1). If not feasible for this publication, it could be a good target for future studies.
The reviewer is right, thank you very much for the correction. We have modified the text accordingly to properly assign lae1 Triha:85012. With respect to paralogs of lae1 Triha:85012, we have not found any protein with a very high similarity to Triha:85012 to suggest that there has been a recent duplication in the genome. We focused on lae1 because of its regulatory role previously described by Aghcheh et al. (2013). However, the function of the other potential methyltransferases cannot be ruled out and therefore might need to be the target for future experiments.
We have rephrased the Discussion section in the manuscript accordingly (New Lines 448-450): “The lae1 gene looks to be a master regulator that is quite conserved and expanded in the genus Trichoderma, but the function of the other potential methyltransferases cannot be ruled out and therefore might need to be the target for future experiments.”
Round 2
Reviewer 1 Report
The authors improved the manuscript. Most of my comments are included in the revision.